# Learning Hippo: A Hippocampal Neural Architecture for Deep Learning

## Abstract

Current neural networks exhibit known limitations due to their fragility against corrupted data and the requirement for large amounts of data for training. These limitations are in deep contrast to the efficiency and robustness of many biological systems such as the human brain. In this work, we present *Hippo-1*, a biological neuron model along with a specific architecture inspired by the CA3 region of the hippocampus. Our biological model is based on a justified compartmental simplification of the neuron which allows us to preserve essential non-linear dendritic integration and on a specific interaction between excitatory and inhibitory neuron populations through a shunting inhibition mechanism. To validate the approach, we evaluated the performance of three simple *Hippo-1* models (named High, Medium, and Low) on standard image classification tasks (i.e., MNIST, Fashion-MNIST, CIFAR-10, CIFAR-100, Tiny ImageNet), comparing them with Multi-Layer Perceptron (MLP) networks of comparable parametric complexity. The experimental results demonstrate that *Hippo-1* not only achieves competitive or superior performance compared to the standard MLPs but also shows a superior robustness to noise (Gaussian and salt-and-pepper), with a more gradual performance degradation as input corruption increases. Gains in adopting Hippo-1 models increase with the complexity of the task, being approximately 5% on Cifar-10, around 10% on Cifar-100 and as high as 50% on Tiny-ImageNet. In fact, our work shows that integrating targeted neuroscientific principles, rather than a complete morphological simulation, represents an effective strategy for building more resilient models, combining biological plausibility with computational efficiency.

## 1 Introduction

Although research in the field of Deep Learning has achieved remarkable results, the current approach has intrinsic limitations that are difficult to resolve. In fact, current models remain on one hand "data-hungry", requiring large amounts of data to generalize, while on the other hand they remain fragile and poorly robust when faced with corrupted or adversarial inputs. This occurs in clear contrast to biological systems, such as the human brain, that demonstrate an amazing capacity for rapid, robust, and energy-efficient learning. This discrepancy suggests a large potential for improvement for biologically inspired approaches especially in what current computational abstractions might be overlooking. Among the most evident simplifications is that of Multi-Layer Perceptrons (MLPs), which, by treating the neuron as a simple linear summator followed by an activation function, represents an excessive simplification of the neuron. In particular, MLPs ignore three fundamental principles of biological neural computation: first, the existence of distinct populations of excitatory (E) and inhibitory (I) neurons, whose dynamic interaction regulates network activity; second, the dendritic morphology and the localized topography of excitatory and inhibitory inputs that enable individual neurons to perform a logical form of dendritic computation; third, the importance of dendritic integration which is intrinsically non-linear, whereby a single neuron processes information in a complex manner in its various dendritic compartments even before reaching the soma (and therefore before generating an output from a potential activation function). Until now, research in this direction has remained constrained by two factors: on one hand, neuroscientific models, while being biologically plausible, are too complex and computationally expensive to be trained at a large scale; on the other hand, machine learning models, although simplistic, have proven so efficient as to make recourse to neuroscientific refinements superfluous and even counterproduc-

tive. The present work is positioned within this context. Instead of simulating the overwhelming morphological complexity of real neurons, in this study we model neurons with a limited number of functional compartments (e.g., apical/basal dendrites, soma), thus preserving essential dendritic computations and E/I interactions, but keeping the model computationally tractable for training on modern hardware. The result is the demonstration not only that it is possible to incorporate some complex neuroscientific principles, such as dendritic integration and shunting inhibition, while remaining computationally efficient and competitive with standard models, but also that such models can provide advantages over usual models. The *Hippo-1* architecture, which we present in this article, explicitly models the interaction between excitatory and inhibitory neurons and embodies a functional representation of dendritic compartments inspired by the CA3 region of the hippocampus. The hippocampus, and in particular the CA3 region, constitutes an ideal field of study for developing more robust and resilient networks due to its recurrent connectivity between pyramidal neurons, specialized in pattern completion, which allows it to reconstruct the correct signal from partial or noise-corrupted inputs. Indeed, the dynamic interaction between excitatory and inhibitory neurons acts as a gain control that stabilizes network activity, making it less susceptible to random perturbations introduced by noise. As a result, the biologically inspired architecture possesses markedly superior noise robustness compared to standard MLP architectures of comparable complexity (see Section 5).

The article is structured as follows: Section 2 analyzes the neuroscientific background that justifies our architectural choices. Section 3 describes the *Hippo-1* model in detail. Section 4 presents our experimental methodology and results obtained on standard datasets. Finally, Sections 5 and 6 critically discuss the results and draw a first round of conclusions on our approach.

## 2 SCIENTIFIC BACKGROUND AND MOTIVATIONS

### 2.1 THE CA3 REGION AS A BIOLOGICAL AUTOASSOCIATIVE NETWORK

In the brain, anatomical connections of different neuron assemblies (groups, nuclei, regions) create the substrate for the function of specific pathways. The entorhinal cortex – hippocampal circuitry has received much attention for its intrinsic nature of comparator Vinogradova (2001) which is crucial for sensory and contextual processing and memory trace formation. The information flow conveyed from various brain areas through the entorhinal cortex to the hippocampal formation is processed in sequence by the different hippocampal regions, i.e., the dentate gyrus, CA3, CA2, and CA1, each endowed with specific "computational" properties. Indeed, the dentate gyrus excels in pattern separation thanks to its extremely sparse coding, but its predominantly feed-forward architecture almost entirely limits its pattern completion capability O'Reilly & McClelland (1994). The CA1 region, on the other hand, functions primarily as an output stage, integrating signals from CA3, CA2 and the entorhinal cortex Bartesaghi et al. (2006), but possesses a sparsely connected recurrent network Fink et al. (2007) and therefore poor autoassociative capabilities. In contrast to the crucial, but somehow "simple", computational tasks accomplished by these hippocampal regions, the CA3 region stands out within the entorhinal cortex – hippocampal circuit as the predominant autoassociative network, due to its extensive recurrent connectivity. Notably, each CA3 pyramidal neuron projects a significant portion of its axon (approximately 30%) to form synapses with other neurons within the same area, creating a densely interconnected architecture Le Duigou et al. (2014). This network of recurrent collaterals forms a dynamic attractor, capable of implementing pattern completion: reactivation of a fraction of a neural pattern (estimated between 30-40%) is sufficient to trigger reactivation of the entire previously memorized activity pattern Rolls (2013). This anatomical property makes this region an almost ideal hub for processing the information conveyed by the two major excitatory input pathways to CA3: the mossy fibers from the dentate gyrus which form extremely powerful, but sparse, "detonator" synapses capable of driving postsynaptic neuron activity with high reliability Henze et al. (2002) and the perforant path from the entorhinal cortex, which provides a direct cortical input, carrying sensory and contextual information Witter (2007). This triple convergence of inputs (recurrent, from dentate gyrus and cortical) confers unique computational richness to the CA3 region. From a functional perspective, this architecture supports several key capabilities. Beyond the aforementioned pattern completion, CA3 is capable of performing pattern separation, transforming similar inputs into more distinct neural representations through competitive dynamics and sparse coding Lee et al. (2015); Knierim & Neunuebel (2016). The plasticity of mossy fiber synapses, moreover, enables rapid learning, sometimes in a single episode (one-shot

learning) Nakazawa et al. (2004). Finally, its recurrent nature makes it suitable for learning temporal sequences Levy (1996). This combination of characteristics has inspired numerous computational models, from Hopfield networks, which abstracted its autoassociative principle Hopfield (1982), to large-scale spiking models that replicate its oscillatory dynamics Kopsick et al. (2023). The CA3 network therefore represents the best compromise, offering a system capable of performing both separation and completion, making it the ideal candidate for a general associative memory model.

## 2.2 FUNCTIONAL ANATOMY AND JUSTIFICATION FOR COMPARTMENTAL SIMPLIFICATION

The remarkable computational capability of CA3 pyramidal neurons emerges from a finely regulated anatomical organization, where synaptic inputs are precisely distributed across different neuronal compartments. Indeed, CA3 pyramidal neurons possess a very complex dendritic morphology, subdivided into distinct functional domains, each receiving specific inputs (Traub et al. (1991); Ascoli et al. (2009)). Thus, signal integration is performed at apical dendrites that receive (from most distal to proximal perisomatic arborization): i) perforant path inputs from the entorhinal cortex; ii) recurrent collaterals from other CA3 neurons and iii) the powerful mossy fiber synapses on the proximal apical dendrites. Similarly, on the basal pole of pyramidal neurons, the basal dendrites integrate additional recurrent inputs and commissural connections with the CA3 region of the contralateral hippocampus. This input segregation is a fundamental computational principle that allows individual neurons to perform a form of dendritic computation (Stingl et al. (2024)). In Figure 1 a schematized pyramidal neuron is shown and the simplified arborization of the four principal dendritic compartments is indicated. Furthermore, inhibitory innervation from different types of GABAergic interneurons, each targeting specific compartments of the dendritic and somatic regions of CA3 pyramidal neurons, implements a system of compartmental inhibition that enables targeted gain control and stabilizes network dynamics (Pelkey et al. (2017)).

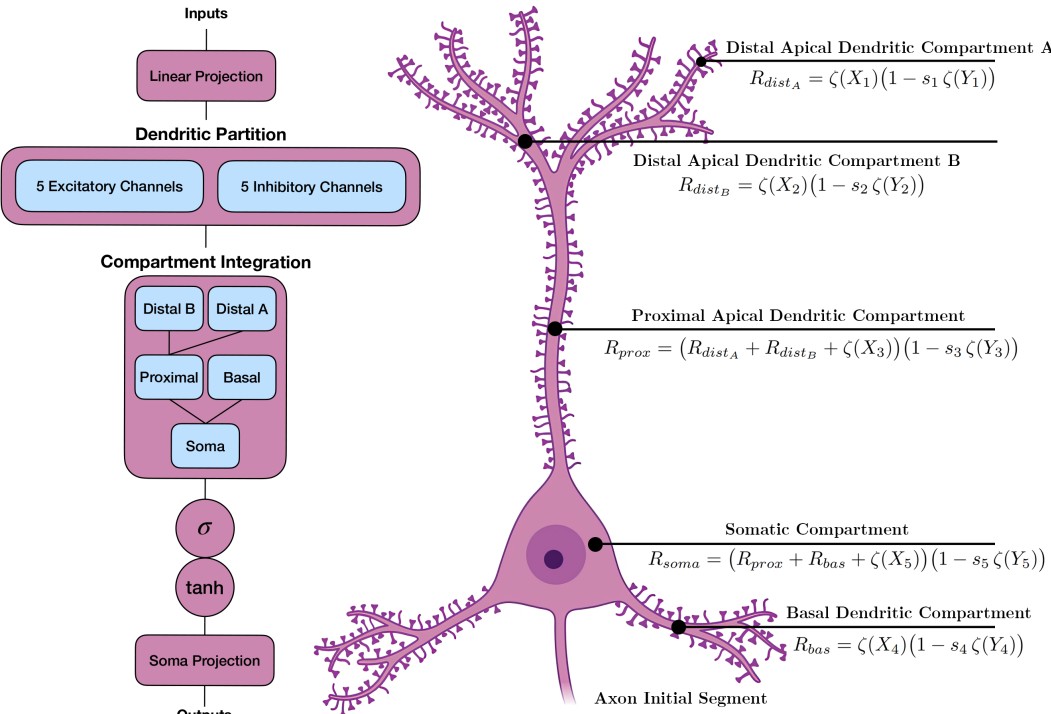

Figure 1: Representation of the five-compartment model for a *Hippo-1* pyramidal neuron with its respective signal integration equations. Each equation incorporates the multiplicative shunting inhibition mechanism where an excitatory signal $X_i$ is multiplicatively modulated by an inhibitory signal $Y_i$.

**A Justified Simplification.** Simulating the complete morphology of a CA3 neuron is computationally prohibitive for large-scale networks. Our strategy is based on a scientifically justified com-

partmental simplification. From an efficiency perspective, reducing a neuron to a few compartments can accelerate simulations by orders of magnitude (Wybo et al. (2021); Hendrickson et al. (2011)). From a biological perspective, this reduction is supported by several principles: i) dendrites organize into functional domains with homogeneous inputs; ii) cable theory demonstrates that electrotonic structure can be approximated by a few equivalent cylinders (Rall (1962); Holmes et al. (1992)); iii) importantly, mechanisms such as "synaptic democracy" (Häusser (2001)) compensate for signal attenuation along the dendrites, making the impact of a given dendritic input less dependent on its distance from the soma (Magee & Cook (2000)). Modeling studies have confirmed that reduced models can faithfully replicate key responses of their morphologically detailed counterparts, and that networks composed of such neurons are capable of reproducing the emergent phenomena of the CA3 region (Kopsick et al. (2023)).

**Proposed Simplified Architecture.** In light of these considerations, we propose an architecture that models each pyramidal neuron with 5 compartments, corresponding to key functional domains (see Figure 1): three apical dendritic domains (two distal and one proximal), one basal dendritic domain, and the soma, from which the axon initial segment detects the global integrated excitation and eventually triggers the action potential. A similar simplified architecture has been adopted for interneurons. This approach represents an optimal equilibrium point as it is complex enough to preserve the fundamental principles of input segregation and non-linear dendritic computation, but simple enough to enable efficient simulations. We argue that the computational power of the CA3 network does not only derive from the fine morphological complexity of pyramidal neurons, but mostly from the organization of information flow, recurrent connectivity, and E/I balance, all principles that our architecture endeavors to preserve.

## 3 THE *Hippo-1* ARCHITECTURE

The architecture of *Hippo-1* was designed to incorporate the CA3-inspired elements described above, while remaining simple enough to be trained with common deep learning techniques. Broadly, the model consists of a multi-stage hierarchical structure where bio-inspired neural modules (imitating excitatory and inhibitory neurons with interaction mechanisms analogous to biological ones) and standard deep learning modules coexist. The specific model used operates through three main phases: an initial input processing stage, a bio-inspired core, potentially organized in multiple stages and a final classification stage. The initial input processing simply consists of a single linear layer that projects the raw data into a higher-dimensional latent space suitable as input for the bio-inspired core. Similarly, the final processing is handled by a traditional dense layer with an activation function for classification, e.g., softmax.

The actual core of the architecture is composed of sequential bio-inspired layers. Each layer contains two parallel modules: one excitatory and one inhibitory, corresponding respectively to the population of pyramidal neurons and GABAergic interneurons of the CA3 region. The goal, as previously mentioned, is to replicate, in a simplified form, the principles of dendritic integration and modulation via shunting inhibition.

Within each module (both excitatory and inhibitory), the input tensor $H \in \mathbb{R}^d$ is divided into five compartments: three apical dendritic compartments (two distal and one proximal); one basal dendritic compartment; and finally, a somatic compartment. In these compartments, the excitatory channels are denoted by $(X_1, \ldots, X_5)$ and the inhibitory ones by $(Y_1, \ldots, Y_5)$.

The signals of the excitatory channels are activated via a non-linear function $\zeta(\cdot)$ (typically `softplus`), while the inhibitory signals are transformed into a suppressive signal (e.g., $-\zeta(\cdot)$). The key interaction between excitatory and inhibitory channels occurs through a multiplicative shunting inhibition mechanism, which modulates the gain of the excitatory signals.

The integration in the various compartments is represented by the responses $R_{dist_A}$, $R_{dist_B}$, $R_{prox}$, $R_{bas}$, and $R_{soma}$, given by the following shunting formulas:

$$R_{dist_A} = \zeta(X_1)\big(1 - s_1\,\zeta(Y_1)\big), \tag{1}$$

$$R_{dist_B} = \zeta(X_2)\big(1 - s_2\,\zeta(Y_2)\big), \tag{2}$$

$$R_{prox} = \big(R_{dist_A} + R_{dist_B} + \zeta(X_3)\big)\big(1 - s_3\,\zeta(Y_3)\big), \tag{3}$$

$$R_{bas} = \zeta(X_4)\big(1 - s_4\,\zeta(Y_4)\big), \tag{4}$$

$$R_{soma} = \big(R_{prox} + R_{bas} + \zeta(X_5)\big)\big(1 - s_5\,\zeta(Y_5)\big), \tag{5}$$

where $s_i \in [0,1]$ is a learned shunting coefficient that determines the intensity of the inhibition. The aggregated output from all compartments is finally processed by a sigmoid function to generate the neuron's output.

The compartmental structure described above is identical for both modules (excitatory and inhibitory) except for an attenuation coefficient for the inhibitory action $\gamma$, which has its reasons in the biology of inhibitory action. Finally, the outputs of the two modules are concatenated, i.e.,

$$H_{\text{out}} = [y_E \;\|\; y_I],$$

so as to allow subsequent layers to learn non-linear combinations of the excitatory and inhibitory contributions.

As can be seen, unlike a standard MLP, each layer of *Hippo-1* implements a functional decomposition into compartments with an interactive non-linearity given by multiplicative shunting and a clear separation between excitatory and inhibitory pathways with dedicated parameters.

## 4 EXPERIMENTAL METHODOLOGY

To evaluate the proposed *Hippo-1* model, we conducted a series of experiments using the `MNIST`, `Fashion MNIST`, `CIFAR-10`, `CIFAR-100` and `Tiny ImageNet` image classification datasets, measuring performance in terms of accuracy, F1-Score and robustness to noise. The idea of using `MNIST` is that it is a relatively simple problem and provides a standard testbed to assess whether a neural architecture can learn effectively and achieve good generalization performance. To our surprise, the *Hippo-1* models not only proved to be capable of learning on `MNIST` comparably to MLPs, but proved to be even better than MLP when the complexity of the task increased as in `CIFAR-10`, `CIFAR-100` and `Tiny ImageNet`. The natural next step was to measure the network's robustness and resilience to data contamination. To do this, we considered two classic types of noise: additive Gaussian noise and salt-and-pepper noise. As hoped, we observed significantly better behavior from the *Hippo-1* models compared to MLPs as discussed in Section 5.

### 4.1 MLP BASELINE

The *Hippo-1* module can be seamlessly integrated with components such as convolutional layers or attention heads, and it scales effectively to deeper architectures. Hippo was conceived as a paradigm shift from MLPs, being more bio-inspired in its design. For this reason, we consider comparisons with MLPs and their capabilities to be the most appropriate. Extensions to more complex architectures are left for future work. We thus defined three fundamental models for Hippo-1 that we named High, Medium and Low (see Table 1) with respect to the number of parameters involved, then we collected the reported accuracy results for a baseline composed by three MLP architecture that had roughly the same amount of learnable parameters: an MLP with one hidden layer of 32 neurons; an MLP with one hidden layer of 64 neurons; an MLP with one hidden layer of 128 neurons. The results for all the models, considered with respect of the number of learnable parameters are in Tables 7 to 4 .

### 4.2 MODEL TRAINING AND OPTIMIZATION

The training of *Hippo-1* was performed in a `JAX/Flax` environment with GPU acceleration. The loss function to minimize was set as the classic *cross-entropy*. For all datasets, we trained each

model with a maximum number of epochs of 500 and terminated via "early stopping" when a new maximum was not reached in 20 new epochs. Training on all datasets was done using stochastic gradient descent with a mini-batch size of 64. As an optimization algorithm, we used the **Adam** optimizer with an adaptive learning rate. To improve convergence, a learning rate schedule with an initial warm-up and subsequent cosine decay was used. Given the non-triviality of the architecture, we adopted a Bayesian hyperparameter optimization approach using the `Optuna` library. Specifically, we defined a search space for some of the main hyperparameters: output dimension per head; hidden dimension $h$ and $c$ of the bio-inspired modules; the shunt coefficient $\gamma$ with continuous values in the range $[0.001, 0.1]$ to modulate the relative strength of inhibition. After about 50 Bayesian search trials, Optuna identified several promising candidate configurations of which we chose three called High, Medium and Low with respect to the number of learnable parameters.

Table 1: Configurations of the three Hippo-1 models (High, Medium, Low). The parameters are as following: `d_out` is the output dimensionality of the initial linear layer; `hidden_dim` is the dimensionality of E/I neurons; `gamma` is the coefficient for shunting inhibition; `n_stages` indicates the number of sequential bio-inspired layers which at this stage of experimentation was set to one, leaving the multistage models for future investigations.

| model_name | d_out | hidden_dim | gamma | n_stages |
|---|---|---|---|---|
| Hippo-1 High | 64 | 128 | 0.001 | 1 |
| Hippo-1 Medium | 32 | 32 | 0.005 | 1 |
| Hippo-1 Low | 16 | 32 | 0.001 | 1 |

To account for possible fluctuations due to random initialization and the order of batch presentation, we retrained each model for 10 independent runs with different initializations. This allowed us to estimate the variability of performance and obtain measures such as the mean and standard deviation of accuracy and other metrics over multiple runs (see Table 7).

### 4.3 NOISE ROBUSTNESS TESTING

Given the biological inspiration of the network, in addition to accuracy measures under normal conditions, we wanted to explore the robustness and resilience of the *Hippo-1* models when faced with corrupted or noise-degraded input data. We considered two common types of noise: Gaussian noise and "salt-and-pepper" noise. For Gaussian noise, we added a random term drawn from a zero-mean normal distribution with variance $\sigma^2$ to each pixel of the test images, with $\sigma = 0.10, 0.20, 0.30, 0.40, 0.50$ (assuming pixels are normalized in [0,1], $\sigma = 0.5$ represents very strong noise that almost completely degrades the visual information). For each value of $\sigma$, we generated a new set of noisy test images and evaluated the model's accuracy and F1-Score performance on them. For "salt-and-pepper" noise, we corrupted a certain percentage $p$ of random pixels by setting them to the maximum (salt, white) or minimum (pepper, black) value with equal probability. We tested $p = 0.10, 0.20, 0.30, 0.40, 0.50$, which corresponds to 10% to 50% of corrupted pixels. Again, for each $p$, we generated noisy versions of the test set and evaluated the performance metrics.

## 5 DISCUSSION OF THE RESULTS

The analysis of the experimental results reveals some interesting features of the *Hippo-1* architecture. As mentioned in the introduction, the experiments were designed to see if it is possible to integrate neuroscientific principles (such as dendritic integration and shunting inhibition) while maintaining competitive performance with standard architectures like MLPs. As a result of our experiments, we can say that the answer to this question is certainly positive.

On the `MNIST` dataset, the *Hippo-1* model achieves performance that is statistically indistinguishable from a comparably sized MLP, reaching an accuracy of 97.78% versus the MLP-128's 97.82% (see Table 7). This negligible difference indicates that the introduction of biologically inspired mechanisms does not penalize learning ability on simple tasks. However, as task complexity increases, a clear advantage for *Hippo-1* emerges. On `CIFAR-10`, the high-parameter version of our model outperforms its MLP counterpart (51.32% vs. 50.53%), and this gap widens significantly on `CIFAR-100`, where *Hippo-1* achieves 24.15% accuracy compared to the MLP's 21.77% (see Tables 5 and 4). The difference becomes even more pronounced on `Tiny ImageNet`, where our

324
325
326
327
328
329
330
331
332
333
334
335
336
337
338
339
340
341

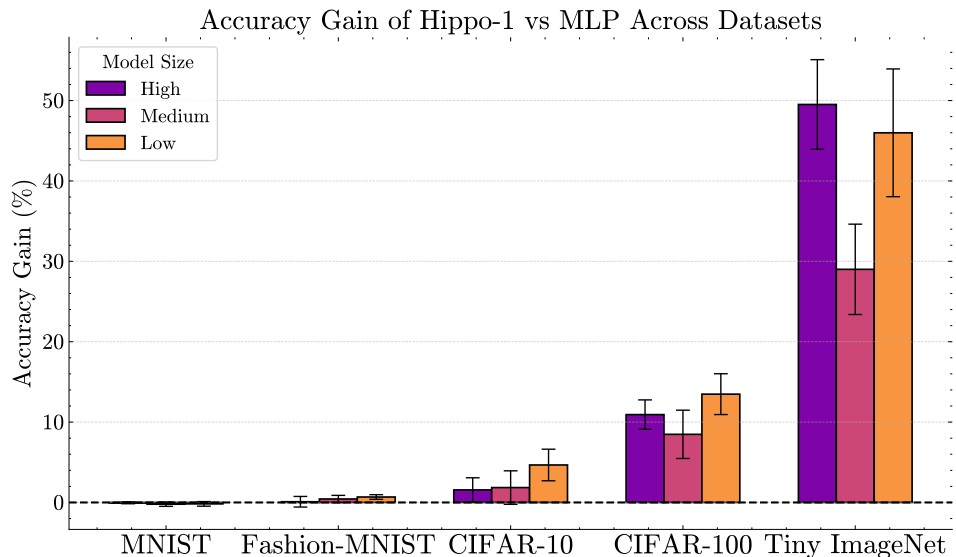

342
343
344
345
346
347
348

Figure 2: Performance improvement of the Hippo-1 models over the MLP baseline across all datasets (see also Tables 7,6, 5, 4 in Appendix). While on easy tasks Hippo models are in par with MLP, the accuracy gain increases with the complexity of the task, being approximately 5% on Cifar-10, around 10% on Cifar-100 and as high as 50% on Tiny-ImageNet.

349
350
351

model delivers at least a 30% relative gain over the MLP baseline. This suggests that the architecture's inherent structure provides a beneficial inductive bias for more challenging problems (see the performance improvement of the Hippo-1 models in Fig. 2).

352
353
354
355
356
357
358
359
360
361
362

It should be noted that *Hippo-1* achieves these results with remarkable parameter efficiency. On `CIFAR-100`, the "Low" variant of *Hippo-1* (with 76k parameters) achieves 21.64% accuracy, decisively outperforming the MLP-32 (with 101k parameters), which only scores 19.07%. This suggests that the proposed architecture can represent information efficiently, achieving high performance even in more compact configurations. Even more so if we consider the Accuracy for MFLOP of the Hippo models compared to its MLP counterpart on the `Tiny ImageNet` (see Table 2). Hippo models are not only significantly more parameter-efficient (e.g., Hippo-Low uses 40.8% fewer parameters than MLP-32) but also require fewer computational resources (40.0% fewer FLOPs). For example, the Hippo-Low model achieves a +145% gain in Accuracy per MFLOP compared to the MLP-32. This finding provides a quantitative validation that our compartmental simplification successfully preserves complex computational capabilities while remaining efficient and tractable on modern hardware.

363
364
365
366
367
368
369
370
371

If the previous results show that the added complexity is not an obstacle but a viable way to build more structured models, the results related to the robustness and resilience of the models in the face of noise-corrupted data are decidedly more interesting. The data show a clear and consistent trend: *Hippo-1* models exhibit a more gradual degradation in performance compared to MLPs. This advantage is particularly pronounced on the more complex datasets. For example, on `CIFAR-100` with very strong Gaussian noise ($\sigma^2$=0.5), *Hippo-1* retains 15.46% accuracy, nearly double the 8.00% of the MLP (see Fig. 3a). The difference becomes even starker with 50% salt-and-pepper noise, where the MLP collapses to 4.70% accuracy (approaching random chance), while *Hippo-1* maintains a score of 11.45%.

372
373
374
375
376
377

This supports the hypothesis that the "shunting inhibition" mechanism acts as a form of dynamic gain control, stabilizing the network's activity and making it less sensitive to random perturbations. The interaction between excitatory and inhibitory modules, inspired by the feedback circuits of CA3, thus seems to confer an intrinsic resilience that is lacking in simpler MLP architectures. This property is consistent with the role of the hippocampal CA3 circuit, known for being an autoassociative network capable of reconstructing complete patterns from incomplete or noisy inputs. In practice, when the input contains noise, the neurons of the bio-inspired model appear better able to discrim-

inate between signal and noise, leading to more stable outputs and correct classification decisions even when the MLP is misled. While the robustness advantage is not as pronounced on simpler datasets like `Fashion-MNIST`, the overall trend confirms that as noise and task complexity increase, the performance of *Hippo-1* degrades more slowly.

Even more interesting results were achieved when the models were trained with a 25%, 50%, 75% and 90% reduction of the datasets. Also in this setting we experienced the trend of increasing gains with higher complexity tasks (see Fig 4). This finding is particularly significant for practical applications, as it suggests that bio-inspired neural architectures can deliver superior performance per computational unit invested, especially for challenging real-world tasks.

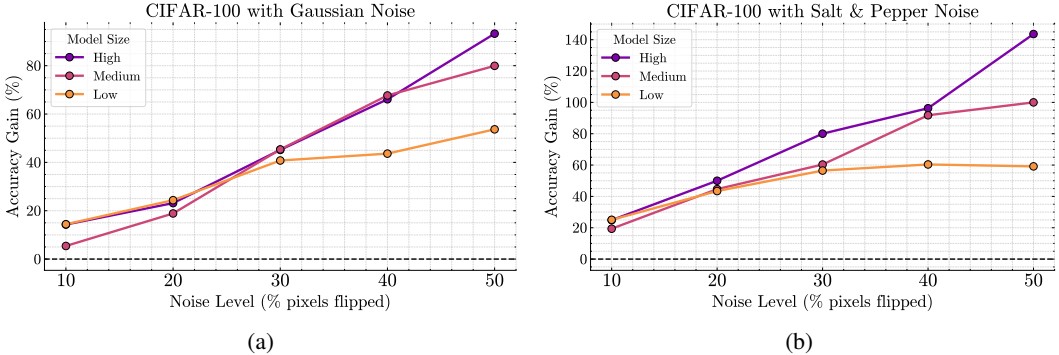

(a)                                    (b)

Figure 3: Performance improvement of the Hippo-1 models over the MLP baseline on `CIFAR-100` when subjected to increasing levels of noise (gaussian, Salt & Pepper. The three lines show the performance of the Hippo-1 Low, Medium, and High models relative to their respective MLP baselines (MLP 32, 64, and 128). A positive value indicates that Hippo-1 performs better than the MLP baseline at that noise level.

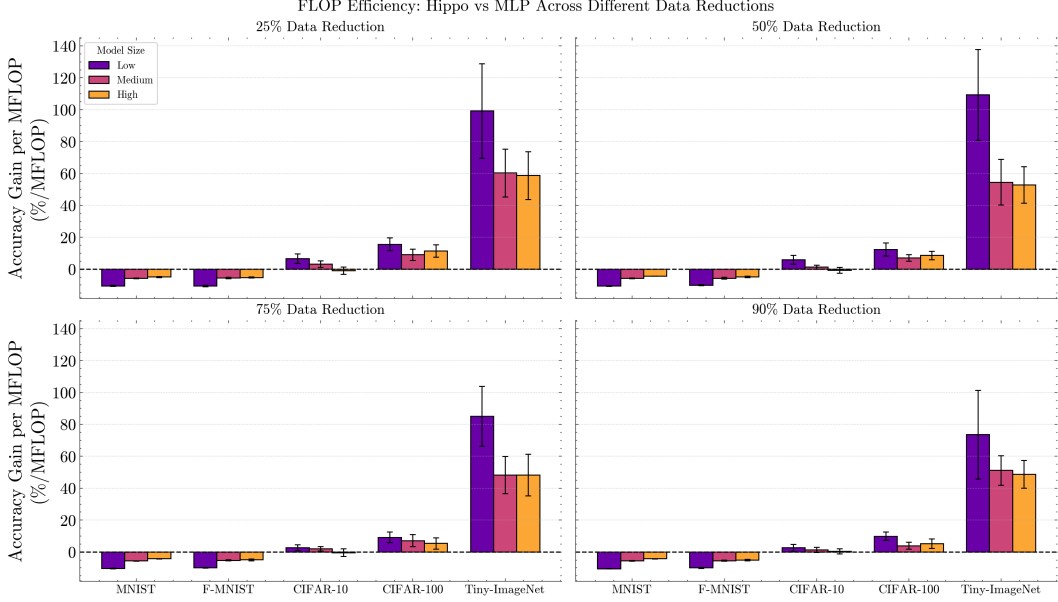

Figure 4: Accuracy gain per MFLOP (%) of Hippo-1 models (Low, Medium, High) compared to parameter-matched MLP baselines across five datasets and four data reduction levels (25%, 50%, 75%, 90% of original training data). Positive values indicate superior computational efficiency of Hippo-1 models. The Hippo-Low configuration consistently exhibits the highest efficiency gains, reaching even 100% improvement on Tiny-ImageNet. Error bars represent standard deviation across multiple independent runs.

| Metric | Hippo-1 | MLP | Winner | % Gain |
|---|---|---|---|---|
| **hippo_low vs fc_32** | | | | |
| Accuracy (mean $\pm$ std) | **$0.0892 \pm 0.0026$** | $0.0611 \pm 0.0041$ | hippo_low | +46.0% |
| F1 Score (mean $\pm$ std) | **$0.0024 \pm 0.0001$** | $0.0016 \pm 0.0001$ | hippo_low | +50.0% |
| Parameters | **236,770** | 399,848 | hippo_low | -40.8% fewer params |
| Accuracy per MFLOP | **0.00291** | 0.00119 | hippo_low | +145% |
| **hippo_medium vs fc_64** | | | | |
| Accuracy (mean $\pm$ std) | **$0.0894 \pm 0.0029$** | $0.0693 \pm 0.0026$ | hippo_medium | +29.0% |
| F1 Score (mean $\pm$ std) | **$0.0024 \pm 0.0001$** | $0.0019 \pm 0.0001$ | hippo_medium | +26.3% |
| Parameters | **443,634** | 799,496 | hippo_medium | -44.5% fewer params |
| Accuracy per MFLOP | **0.00157** | 0.000678 | hippo_medium | +131% |
| **hippo_high vs fc_128** | | | | |
| Accuracy (mean $\pm$ std) | **$0.1081 \pm 0.0029$** | $0.0723 \pm 0.0028$ | hippo_high | +49.6% |
| F1 Score (mean $\pm$ std) | **$0.0029 \pm 0.0001$** | $0.0020 \pm 0.0001$ | hippo_high | +45.0% |
| Parameters | **1,118,482** | 1,598,792 | hippo_high | -30.0% fewer params |
| Accuracy per MFLOP | **0.000748** | 0.000354 | hippo_high | +111% |

Table 2: Comparison between Hippo models and MLP on the TinyImageNet dataset (see Appendix for all metrics).

## 6 CONCLUSIONS AND FUTURE WORK

In this work, we introduced Hippo-1, a neural network architecture that integrates fundamental computational principles of the hippocampal CA3 region, including the segregation of inputs into functional compartments and the dynamic interaction between excitatory and inhibitory neurons through shunting inhibition. Our research has shown that this approach achieves a dual objective: it maintains competitive performance compared to standard models like MLPs on image classification tasks, and, more importantly, it exhibits significantly superior robustness in the presence of noise. This advantage becomes particularly evident on more challenging benchmarks such as `Tiny ImageNet` (on which Hippo models obtained as much as 145% gain on Accuracy per MFLOP), further reinforcing the scalability and its potential applicability to larger, real-world datasets. This validates our thesis for which biological plausibility, when implemented through targeted computational principles rather than a complete morphological simulation, is not just a theoretical exercise but proves to be a practical strategy for building more resilient deep learning models. The success of *Hippo-1* opens the door to numerous future research directions. A natural first step will be to extend the architecture to more faithfully model the recurrent connectivity of CA3 and to integrate the dentate gyrus. Implementing explicit recurrent connections within the excitatory modules could unlock the full autoassociative capabilities of the model, allowing us to test it on tasks for which CA3 is biologically specialized, such as pattern completion from partial inputs, one-shot learning, and temporal sequence memorization. In parallel, a dedicated study should be considered for the network's training method, which currently lacks any biological support or analogy. Finally, exploring more biologically plausible learning rules, such as forms of Hebbian plasticity, could further reduce the reliance on large amounts of labeled data and bring the model closer to the efficient learning capabilities of the brain. We believe that this hybrid approach, which merges the efficiency of modern deep learning algorithms with the robust principles discovered by neuroscience, represents a fertile path for the development of the next generation of artificial intelligence.

### REPRODUCIBILITY STATEMENT

To ensure the reproducibility of our research, a comprehensive explanation of the *Hippo-1* architecture is provided in Section 3. The complete experimental methodology, including details on the datasets, baseline models, training procedures, and hyperparameter optimization, is described in Section 4 and raw result files from all experimental runs are available for independent verification of the performance metrics reported at `https://Undisclosed_for_Anonymity_Requirements`.

## Use of Large Language Models

Large Language Models were used to translate and polish the work in english.

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

## A  Appendix

You may include other additional sections here.

| Metric | Hippo-1 | MLP | Winner | % Gain |
|---|---|---|---|---|
| **hippo_low vs fc_32** | | | | |
| Accuracy (mean ± std) | 0.0892 ± 0.0026 | 0.0611 ± 0.0041 | hippo_low | +46.0% |
| F1 Score (mean ± std) | 0.0024 ± 0.0001 | 0.0016 ± 0.0001 | hippo_low | +50.0% |
| Training Time (s) | 978.3 ± 82.1 | 727.8 ± 105.2 | fc_32 | -34.5% (hippo_low slower) |
| Parameters | 236,770 | 399,848 | hippo_low | -40.8% fewer params |
| FLOPs (total) | 30.7 M | 51.2 M | hippo_low | -40.0% fewer FLOPs |
| FLOPs per sample | 0.48 M | 0.80 M | hippo_low | -40.0% fewer FLOPs |
| Accuracy per MFLOP | 0.00291 | 0.00119 | hippo_low | +145% |
| **hippo_medium vs fc_64** | | | | |
| Accuracy (mean ± std) | 0.0894 ± 0.0029 | 0.0693 ± 0.0026 | hippo_medium | +29.0% |
| F1 Score (mean ± std) | 0.0024 ± 0.0001 | 0.0019 ± 0.0001 | hippo_medium | +26.3% |
| Training Time (s) | 1556.9 ± 1473.6 | 679.0 ± 118.0 | fc_64 | -129% (hippo slower) |
| Parameters | 443,634 | 799,496 | hippo_medium | -44.5% fewer params |
| FLOPs (total) | 57.1 M | 102.3 M | hippo_medium | -44.2% fewer FLOPs |
| FLOPs per sample | 0.89 M | 1.60 M | hippo_medium | -44.2% fewer FLOPs |
| Accuracy per MFLOP | 0.00157 | 0.000678 | hippo_medium | +131% |
| **hippo_high vs fc_128** | | | | |
| Accuracy (mean ± std) | 0.1081 ± 0.0029 | 0.0723 ± 0.0028 | hippo_high | +49.6% |
| F1 Score (mean ± std) | 0.0029 ± 0.0001 | 0.0020 ± 0.0001 | hippo_high | +45.0% |
| Training Time (s) | 786.4 ± 38.2 | 589.6 ± 45.7 | fc_128 | -33.4% (hippo_high slower) |
| Parameters | 1,118,482 | 1,598,792 | hippo_high | -30.0% fewer params |
| FLOPs (total) | 144.6 M | 204.6 M | hippo_high | -29.3% fewer FLOPs |
| FLOPs per sample | 2.26 M | 3.20 M | hippo_high | -29.3% fewer FLOPs |
| Accuracy per MFLOP | 0.000748 | 0.000354 | hippo_high | +111% |

Table 3: Complete comparison between all models on TinyImageNet

| Metric | Hippo-1 | MLP | Winner | % Gain |
|---|---|---|---|---|
| **hippo_low vs fc_32** | | | | |
| Accuracy (mean ± std) | 0.2161 ± 0.0029 | 0.1913 ± 0.0049 | hippo_low | +12.9% |
| F1 Score (mean ± std) | 0.0937 ± 0.0011 | 0.0819 ± 0.0026 | hippo_low | +14.4% |
| Training Time (s) | 726.2 ± 88.9 | 704.2 ± 159.1 | fc_32 | -3.1% (hippo slower) |
| Parameters | 76,414 | 101,636 | hippo_low | -24.8% fewer |
| FLOPs (total) | 10.14 M | 12.99 M | hippo_low | -22.0% fewer |
| FLOPs per sample | 158,388 | 203,040 | hippo_low | -22.0% fewer |
| Accuracy per MFLOP | 0.0213 | 0.0147 | hippo_low | +44.9% |
| **hippo_medium vs fc_64** | | | | |
| Accuracy (mean ± std) | 0.2210 ± 0.0024 | 0.2004 ± 0.0061 | hippo_medium | +10.3% |
| F1 Score (mean ± std) | 0.0967 ± 0.0009 | 0.0861 ± 0.0031 | hippo_medium | +12.3% |
| Training Time (s) | 472.3 ± 140.1 | 443.3 ± 51.5 | fc_64 | -6.5% (hippo slower) |
| Parameters | 135,822 | 203,172 | hippo_medium | -33.2% fewer |
| FLOPs (total) | 17.74 M | 25.99 M | hippo_medium | -31.7% fewer |
| FLOPs per sample | 277,188 | 406,080 | hippo_medium | -31.7% fewer |
| Accuracy per MFLOP | 0.0125 | 0.00771 | hippo_medium | +62.0% |
| **hippo_high vs fc_128** | | | | |
| Accuracy (mean ± std) | 0.2412 ± 0.0031 | 0.2179 ± 0.0039 | hippo_high | +10.7% |
| F1 Score (mean ± std) | 0.1050 ± 0.0021 | 0.0945 ± 0.0026 | hippo_high | +11.1% |
| Training Time (s) | 549.0 ± 96.8 | 224.7 ± 52.5 | fc_128 | -144.3% (hippo slower) |
| Parameters | 477,358 | 406,244 | fc_128 | +17.5% more |
| FLOPs (total) | 62.55 M | 51.98 M | fc_128 | +20.3% more |
| FLOPs per sample | 977,316 | 812,160 | fc_128 | +20.3% more |
| Accuracy per MFLOP | 0.00386 | 0.00419 | fc_128 | -7.9% worse |

Table 4: Complete comparison between all models on CIFAR-100

| Metric | Hippo-1 | MLP | Winner | % Gain |
|---|---|---|---|---|
| **hippo_low vs fc_32** | | | | |
| Accuracy (mean ± std) | 0.4902 ± 0.0068 | 0.4644 ± 0.0073 | hippo_low | +5.6% |
| F1 Score (mean ± std) | 0.4652 ± 0.0049 | 0.4389 ± 0.0096 | hippo_low | +6.0% |
| Training Time (s) | 702.6 ± 74.3 | 519.5 ± 71.3 | fc_32 | -35.2% (hippo slower) |
| Parameters | 64,804 | 98,666 | hippo_low | -34.3% fewer |
| FLOPs (total) | 8.66 M | 12.63 M | hippo_low | -31.4% fewer |
| FLOPs per sample | 135,258 | 197,280 | hippo_low | -31.4% fewer |
| Accuracy per MFLOP | 0.00566 | 0.00368 | hippo_low | +53.7% |
| **hippo_medium vs fc_64** | | | | |
| Accuracy (mean ± std) | 0.5016 ± 0.0089 | 0.4860 ± 0.0075 | hippo_medium | +3.2% |
| F1 Score (mean ± std) | 0.4758 ± 0.0098 | 0.4597 ± 0.0074 | hippo_medium | +3.5% |
| Training Time (s) | 436.18 ± 36.91 | 375.43 ± 52.73 | fc_64 | -16.2% (hippo slower) |
| Parameters | 124,212 | 197,322 | hippo_medium | -37.1% fewer |
| FLOPs (total) | 16.26 M | 25.25 M | hippo_medium | -35.6% fewer |
| FLOPs per sample | 254,058 | 394,560 | hippo_medium | -35.6% fewer |
| Accuracy per MFLOP | 0.00308 | 0.00193 | hippo_medium | +59.6% |
| **hippo_high vs fc_128** | | | | |
| Accuracy (mean ± std) | 0.5150 ± 0.0095 | 0.4996 ± 0.0064 | hippo_high | +3.1% |
| F1 Score (mean ± std) | 0.4899 ± 0.0086 | 0.4740 ± 0.0078 | hippo_high | +3.3% |
| Training Time (s) | 590.96 ± 59.68 | 313.26 ± 26.64 | fc_128 | -88.6% (hippo slower) |
| Parameters | 431,188 | 394,634 | fc_128 | +9.3% more |
| FLOPs (total) | 56.64 M | 50.50 M | fc_128 | +12.1% more |
| FLOPs per sample | 885,066 | 789,120 | fc_128 | +12.1% more |
| Accuracy per MFLOP | 0.00909 | 0.00989 | fc_128 | -8.1% worse |

Table 5: Complete comparison between all models on CIFAR-10

| Metric | Hippo-1 | MLP | Winner | % Gain |
|--------|---------|-----|--------|--------|
| **hippo_low vs fc_32** | | | | |
| Accuracy (mean ± std) | 0.8718 ± 0.0037 | 0.8641 ± 0.0034 | hippo_low | +0.9% |
| F1 Score (mean ± std) | 0.8587 ± 0.0043 | 0.8499 ± 0.0038 | hippo_low | +1.0% |
| Training Time (s) | 501.8 ± 93.2 | 348.4 ± 60.8 | fc_32 | -43.9% (hippo slower) |
| Parameters | 28,196 | 25,450 | fc_32 | +10.8% more |
| FLOPs (total) | 3.97 M | 3.25 M | fc_32 | +22.0% more |
| FLOPs per sample | 62,042 | 50,848 | fc_32 | +22.0% more |
| Accuracy per MFLOP | 0.219 | 0.266 | fc_32 | -17.7% worse |
| **hippo_medium vs fc_64** | | | | |
| Accuracy (mean ± std) | 0.8785 ± 0.0030 | 0.8764 ± 0.0059 | hippo_medium | +0.2% |
| F1 Score (mean ± std) | 0.8658 ± 0.0035 | 0.8634 ± 0.0061 | hippo_medium | +0.3% |
| Training Time (s) | 349.8 ± 37.4 | 350.5 ± 35.5 | Tie | -0.2% faster |
| Parameters | 50,996 | 50,890 | Tie | +0.2% more |
| FLOPs (total) | 6.89 M | 6.51 M | fc_64 | +5.8% more |
| FLOPs per sample | 107,626 | 101,696 | fc_64 | +5.8% more |
| Accuracy per MFLOP | 0.128 | 0.135 | fc_64 | -5.2% worse |
| **hippo_high vs fc_128** | | | | |
| Accuracy (mean ± std) | 0.8814 ± 0.0026 | 0.8819 ± 0.0031 | fc_128 | -0.1% worse |
| F1 Score (mean ± std) | 0.8690 ± 0.0022 | 0.8696 ± 0.0037 | fc_128 | -0.1% worse |
| Training Time (s) | 683.3 ± 60.6 | 379.4 ± 52.9 | fc_128 | +80% (hippo slower) |
| Parameters | 284,756 | 101,770 | fc_128 | +180% more |
| FLOPs (total) | 37.9 M | 13.0 M | fc_128 | +191% more |
| FLOPs per sample | 592,202 | 203,392 | fc_128 | +191% more |
| Accuracy per MFLOP | 0.0233 | 0.0678 | fc_128 | -65.6% worse |

Table 6: Complete comparison between all models on Fashion-MNIST

| Metric | Hippo-1 | MLP | Winner | % Gain |
|--------|---------|-----|--------|--------|
| **hippo_low vs fc_32** | | | | |
| Accuracy (mean ± std) | 0.9653 ± 0.0017 | 0.9667 ± 0.0010 | fc_32 | -0.1% worse |
| F1 Score (mean ± std) | 0.9605 ± 0.0020 | 0.9619 ± 0.0012 | fc_32 | -0.1% worse |
| Training Time (s) | 407.8 ± 49.1 | 367.1 ± 53.9 | fc_32 | +11.1% (hippo slower) |
| Parameters | 28,196 | 25,450 | fc_32 | +10.8% more |
| FLOPs (total) | 3.97 M | 3.25 M | fc_32 | +22.0% more |
| FLOPs per sample | 62,042 | 50,848 | fc_32 | +22.0% more |
| Accuracy per MFLOP | 0.243 | 0.298 | fc_32 | -18.4% worse |
| **hippo_medium vs fc_64** | | | | |
| Accuracy (mean ± std) | 0.9717 ± 0.0017 | 0.9727 ± 0.0010 | fc_64 | -0.1% worse |
| F1 Score (mean ± std) | 0.9673 ± 0.0020 | 0.9685 ± 0.0010 | fc_64 | -0.1% worse |
| Training Time (s) | 347.0 ± 47.8 | 309.5 ± 43.5 | fc_64 | +12.1% (hippo slower) |
| Parameters | 50,996 | 50,890 | Tie | +0.2% more |
| FLOPs (total) | 6.89 M | 6.51 M | fc_64 | +5.8% more |
| FLOPs per sample | 107,626 | 101,696 | fc_64 | +5.8% more |
| Accuracy per MFLOP | 0.141 | 0.149 | fc_64 | -5.5% worse |
| **hippo_high vs fc_128** | | | | |
| Accuracy (mean ± std) | 0.9735 ± 0.0035 | 0.9770 ± 0.0008 | fc_128 | -0.36% worse |
| F1 Score (mean ± std) | 0.9696 ± 0.0036 | 0.9733 ± 0.0009 | fc_128 | -0.38% worse |
| Training Time (s) | 578.4 ± 114.7 | 464.6 ± 85.7 | fc_128 | +24.5% (hippo slower) |
| Parameters | 284,756 | 101,770 | fc_128 | +180% more |
| FLOPs (total) | 37.90 M | 13.02 M | fc_128 | +191% more |
| FLOPs per sample | 592,202 | 203,392 | fc_128 | +191% more |
| Accuracy per MFLOP | 0.0257 | 0.0750 | fc_128 | -65.7% worse |

Table 7: Complete comparison between all models on MNIST

Table 8: Model accuracy on MNIST with Gaussian Noise

| Modello | Var 0.10 | Var 0.20 | Var 0.30 | Var 0.40 | Var 0.50 |
|---|---|---|---|---|---|
| MLP 128 | 0.9501 | 0.7737 | 0.5878 | 0.4474 | 0.3523 |
| **Hippo-1 High** | **0.9551** | **0.7886** | **0.5915** | **0.4623** | **0.3687** |
| MLP 64 | **0.9169** | 0.6898 | 0.4749 | 0.3303 | 0.2524 |
| **Hippo-1 Medium** | 0.9132 | **0.7016** | **0.5064** | **0.3814** | **0.3016** |
| MLP 32 | 0.8100 | 0.5044 | 0.3622 | 0.2750 | 0.2135 |
| **Hippo-1 Low** | **0.7776** | **0.5424** | **0.4045** | **0.3110** | **0.2509** |

Table 9: Model accuracy on MNIST with Salt & Pepper Noise

| Modello | Perc 10% | Perc 20% | Perc 30% | Perc 40% | Perc 50% |
|---|---|---|---|---|---|
| MLP 128 | 0.7465 | 0.5215 | 0.3807 | 0.2909 | 0.2308 |
| **Hippo-1 High** | **0.8092** | **0.5748** | **0.4226** | **0.3248** | **0.2561** |
| MLP 64 | 0.6837 | 0.4456 | 0.3078 | 0.2280 | 0.1794 |
| **Hippo-1 Medium** | **0.7232** | **0.4850** | **0.3548** | **0.2639** | **0.2091** |
| MLP 32 | 0.5703 | 0.3595 | 0.2524 | 0.2008 | 0.1704 |
| **Hippo-1 Low** | **0.6183** | **0.4036** | **0.2966** | **0.2265** | **0.1882** |

Table 10: Model accuracy on Fashion MNIST with Gaussian Noise

| Modello | Var 0.10 | Var 0.20 | Var 0.30 | Var 0.40 | Var 0.50 |
|---|---|---|---|---|---|
| **MLP 128** | **0.8326** | **0.6580** | **0.4909** | **0.3664** | **0.2734** |
| Hippo-1 High | 0.8079 | 0.6282 | 0.4407 | 0.3163 | 0.2397 |
| **MLP 64** | **0.8180** | **0.6513** | **0.4953** | **0.3688** | **0.2908** |
| Hippo-1 Medium | 0.7930 | 0.5742 | 0.3974 | 0.2993 | 0.2358 |
| **MLP 32** | **0.8163** | **0.6946** | **0.5552** | **0.4300** | **0.3415** |
| Hippo-1 Low | 0.7791 | 0.5563 | 0.3879 | 0.2849 | 0.2190 |

Table 11: Model accuracy on Fashion MNIST with Salt & Pepper Noise

| Modello | Perc 10% | Perc 20% | Perc 30% | Perc 40% | Perc 50% |
|---|---|---|---|---|---|
| **MLP 128** | **0.6391** | **0.4387** | **0.3230** | **0.2362** | **0.1998** |
| Hippo-1 High | 0.6583 | 0.4539 | 0.3123 | 0.2216 | 0.1700 |
| **MLP 64** | **0.6296** | **0.4436** | **0.3186** | **0.2451** | **0.1926** |
| Hippo-1 Medium | 0.6283 | 0.4285 | 0.3086 | 0.2295 | 0.1837 |
| **MLP 32** | **0.6712** | **0.5126** | **0.3795** | **0.2918** | **0.2272** |
| Hippo-1 Low | 0.6174 | 0.4229 | 0.2921 | 0.2139 | 0.1799 |

Table 12: Model accuracy on CIFAR-10 with Gaussian Noise

| Modello | Var 0.10 | Var 0.20 | Var 0.30 | Var 0.40 | Var 0.50 |
|---|---|---|---|---|---|
| MLP 128 | 0.4884 | 0.4508 | 0.3990 | 0.3656 | 0.3222 |
| **Hippo-1 High** | **0.5012** | **0.4726** | **0.4553** | **0.4182** | **0.3739** |
| MLP 64 | 0.4743 | 0.4368 | 0.3918 | 0.3538 | 0.3095 |
| **Hippo-1 Medium** | **0.4994** | **0.4813,** | **0.4505** | **0.4163** | **0.3759** |
| MLP 32 | 0.4585 | 0.4224 | 0.3868 | 0.3325 | 0.2910 |
| **Hippo-1 Low** | **0.4823** | **0.4613** | **0.4269** | **0.3791** | **0.3421** |

Table 13: Model accuracy on CIFAR-10 with Salt & Pepper Noise

| Modello | Perc 10% | Perc 20% | Perc 30% | Perc 40% | Perc 50% |
|---|---|---|---|---|---|
| MLP 128 | 0.4610 | 0.4062 | 0.3647 | 0.3021 | 0.2568 |
| **Hippo-1 High** | **0.4843** | **0.4557** | **0.4124** | **0.3649** | **0.3027** |
| MLP 64 | 0.4468 | 0.3982 | 0.3434 | 0.2925 | 0.2487 |
| **Hippo-1 Medium** | **0.4852** | **0.4483** | **0.4078** | **0.3665** | **0.3109** |
| MLP 32 | 0.4298 | 0.3729 | 0.3237 | 0.2831 | 0.2419 |
| **Hippo-1 Low** | **0.4714** | **0.4208** | **0.3740** | **0.3283** | **0.2826** |

Table 14: Model accuracy on CIFAR-100 with Gaussian Noise

| Modello | Var 0.10 | Var 0.20 | Var 0.30 | Var 0.40 | Var 0.50 |
|---|---|---|---|---|---|
| MLP 128 | 0.2072 | 0.1810 | 0.1406 | 0.1062 | 0.0800 |
| **Hippo-1 High** | **0.2368** | **0.2229** | **0.2041** | **0.1764** | **0.1546** |
| MLP 64 | 0.2032 | 0.1664 | 0.1261 | 0.0947 | 0.0713 |
| **Hippo-1 Medium** | **0.2142** | **0.1978** | **0.1833** | **0.1588** | **0.1283** |
| MLP 32 | 0.1946 | 0.1638 | 0.1317 | 0.1066 | 0.0805 |
| **Hippo-1 Low** | **0.2226** | **0.2037** | **0.1854** | **0.1531** | **0.1237** |

Table 15: Model accuracy on CIFAR-100 with Salt & Pepper Noise

| Modello | Perc 10% | Perc 20% | Perc 30% | Perc 40% | Perc 50% |
|---|---|---|---|---|---|
| MLP 128 | 0.1791 | 0.1342 | 0.0991 | 0.0723 | 0.0470 |
| **Hippo-1 High** | **0.2236** | **0.2013** | **0.1784** | **0.1419** | **0.1145** |
| MLP 64 | 0.1698 | 0.1241 | 0.0938 | 0.0608 | 0.0425 |
| **Hippo-1 Medium** | **0.2027** | **0.1795** | **0.1504** | **0.1166** | **0.0850** |
| MLP 32 | 0.1664 | 0.1267 | 0.0963 | 0.0727 | 0.0524 |
| **Hippo-1 Low** | **0.2081** | **0.1817** | **0.1507** | **0.1166** | **0.0834** |

