# OpenReview forum: "Learning Hippo: A Hippocampal Neural Architecture for Deep Learning"
_ICLR.cc/2026/Conference — Submitted to ICLR 2026_

### Official Review · Reviewer_4kxP · 2025-10-27

**Soundness:** 3
**Presentation:** 2
**Contribution:** 2
**Rating:** 4
**Confidence:** 3

**Summary:**

The authors propose a new architecture, inspired by the hippocampus, in which various neurons (or is it neural projections from a given neuron?) are divided into multiple compartments, such that each compartment affects target neurons at a different stage of processing (?).

Based on multiple small-scale image classification experiments, the proposed model seems to outperform simple one-hidden-layer multi-layer perceptrons (MLPs) for a given number of trainable parameters, without increasing computational requirements.

**Strengths:**

The architecture seems novel. The results are intriguing. The review of hippocampal function is informative.

**Weaknesses:**

The main problem is that the architecture is not precisely described, making it impossible to reproduce (or understand), and thus preventing acceptance.

Here are a few questions:

- Where are the weights? The architecture  seems to have some kind of trainable connection weights, judging from the main text. However, no such trainable weights appear in Equations 1-5 - only the inhibitory-specific slope parameters s are shown. As a result it is impossible to know at which of the successive "stages" (if any) trainable weights are applied.

- We are told that there is a "gamma" parameter denoting an "attenuation coefficient" for inhibition. Where is it applied?

- In line 234 we are told that further layers should learn "nonlinear combinations" of the concatenated inhibitory and excitatory portions of H. But in line 210 we are told that the input H is just split into two parts for inhibition and excitation. Which is it?

- Table 1 suggests the initial linear layer has dimension different from the "dimensionality" of E/I neurons. This suggests that some sort of projection occurs between the linear layer and the E/I neurons. But no such projection is reported in the equations, which instead suggest the input is simply partitioned and then a series of nonlinearities is applied to each neuron/channel in parallel.

- More generally it is never clear what counts as a single neuron, a whole group of neurons, or a single channel (a connection between neurons)

- l. 277: "hidden dimensions h and c of the bio-inspired modules" - what are those? They don't appear in the equations.

Basically, the architecture must be explained precisely, *with equations and/or pseudocode* describing the whole process, and clearly defining *all* symbols (and also clearly differentiating scalars, vectors and matrices). At present, the description, such as it is, is not ready for publication.

There are two additional caveats:

- It's not clear how fair the comparison with MLP is. The HIPPO model includes a lot more nonlinearities and multiplicative interactions. It also had its hyperparameters carefully tuned. No such tuning is reported for the MLPs. At the very least, MLPs with two layers (but similar number of overall parameters) should be tried.


- It is interesting that the proposed model does not seem to require more computation (per parameter) than MLPs. However, the specific metric of "accuracy gain per flop" is described as dividing the accuracy gain by the flops, which seems strange (e.g. if they had exact same accuracy, but very different flops, the result would be 0, indicating no difference, which would not reflect the very different efficiencies. Admittedly it is not obvious how to jointly compare two independent dimensions.

Minor:

- Fix the citations, using correct parenthetical style (the authors may find the following latex command helpful: \renewcommand{\cite}{\citep}  )

- l. 271 "when a new maximum was not reached..." - a new maximum of what?

**Questions:**

See above.

---

### Official Review · Reviewer_Vrmw · 2025-11-01

**Soundness:** 1
**Presentation:** 1
**Contribution:** 1
**Rating:** 2
**Confidence:** 4

**Summary:**

This paper aims to propose a biologically-inspired architecture for deep learning, inspired by hippocampal pyramidal neurons, with the goal of improving performance on specific tasks and the robustness of deep learning models to input perturbations. First, the authors propose their core model architecture which involves separate excitatory and inhibitory components/channels which interact through several additive and multiplicative steps, improving the expressivity of the architecture over simple artificial neural network (ANN) units typically found in multi-layer perceptrons (MLPs). Then, the authors evaluate their method on several image classification benchmarks and compare its performance with that of MLPs in terms of classification accuracy and robustness to corruption in input image pixels (with Gaussian or salt-and-pepper noise). The authors find that on more complex classification tasks (CIFAR-100 & TinyImageNet), their method outperforms MLPs while typically using fewer FLOPS overall and also working better with less training data. Overall, the claim is that architectures with a certain degree of bio-inspiration can outperform typical ANNs and serve as a promising replacement for typical deep learning models.

**Strengths:**

* An implementation of biological neural computations in artificial neural network models could improve our understanding of how the brain works and which features of neural computation contribute towards improving performance and robustness.
* The proposed model is more efficient in terms of number of FLOPS (and thus, the amount of compute) and typically has fewer parameters than the chosen competitor model.

**Weaknesses:**

* Why do the authors consider an MLP, that too one with a single hidden layer alone and just different numbers of units, as the only baseline comparison? There are several other architectures, even just MLPs (although CNNs, ViTs, etc. are highly performant) that perform better than the MLP baseline considered here, and are also quite parameter-efficient or at least comparable. In particular, models like simple CNNs could very reasonably outperform Hippo-1 with far fewer parameters (and the CNN has bio-inspired origins too). There is also no comparison to bio-inspired architectures such as spiking MLPs, e.g., https://arxiv.org/abs/2203.14679 or others.
* On a related note, the authors have considered only a single architecture with the MLP as a base and not adapted their model to work with CNN-style layers. This severely limits the applicability of the framework introduced.
* Only image classification tasks have been considered, that too, with an MLP-like architecture. To make a convincing case for a new architecture, I would expect additional tasks (e.g., regression, sequence prediction, etc.) and also for the framework/modifications to be integrated into more modern models and other architectures such as CNNs, RNNs, or Transformers – and to test whether there are actually any meaningful improvements.
* While the authors have tested the robustness of their model to some input perturbations, e.g., Gaussian and salt-and-pepper noise, they have not evaluated whether their models are robust to more sophisticated perturbations such as those generated through adversarial attacks, which can be specifically optimised to increase misclassification in a model-specific way. Claims about improved robustness would, in my opinion, require testing against sophisticated threats that deep learning models could face in the real world.
* It is unclear how important the several hyperparameters are to the Hippo-1 architecture, as there is no reported robustness study to changes in hyperparameter values. Furthermore, it does not seem like an equivalent hyperparameter optimisation scheme was considered for the MLP baseline, which calls into question the fairness of the comparisons reported here.
* A critical weakness is that the actual accuracy values reported, for both the MLP and Hippo-1, are extremely low in comparison to typical model accuracies on these datasets. In fact, I have in my experience trained simple MLPs that perform quite a bit better than the reported baseline MLP on datasets such as CIFAR-10. To be precise, the reported accuracies for Hippo-1 on CIFAR-10, CIFAR-100 are 51.5%, 24.1% – these are really low, vanilla fully-connected networks have been shown to perform far better (https://arxiv.org/pdf/2007.13657).
* There is no discussion of a large amount of relevant work on bio-inspired architectures for deep learning and image classification. I am not providing an exhaustive list here but references to fundamental papers on CNNs, the dataset publications, papers on gating and other mechanisms added to ANNs and inspired by the brain, etc. are missing.
* Implementation details are very sparse and I am unable to get a good understanding of the architecture from Section 3. There is also no code provided.
* Finally, there are several formatting and presentation issues, the first of which could be quite serious in my opinion:
    * There are 3 garbled references with incorrect information: (1) "Quantitative morphometry..." by Ascoli et al. should have authors Ascoli, Brown, Calixto, Card, et al. (https://pubmed.ncbi.nlm.nih.gov/19496174/), but the listed authors are those on another paper by Ascoli et al. (https://pubmed.ncbi.nlm.nih.gov/20445069/); (2) "Robust resting-state..." by Kopsick et al. should have authors Kopsick, Tecuati, Moradi, Attili, et al. (https://pubmed.ncbi.nlm.nih.gov/37663748/), but the listed authors are from another Kopsick et al. paper (https://link.springer.com/article/10.1007/s10827-024-00881-3); (3) Stingl, Draguhn & Both has incorrect issue information (https://pubmed.ncbi.nlm.nih.gov/39162833/).
    * Template text has not been removed from the appendix.
    * The paper seems heavily LLM-edited overall – although, the authors acknowledge that LLMs were used to translate the paper into English and polish the text. Typical papers contain several subsections and paragraphs, and references are not just scoped to one literature review section but are spread throughout. Here, the overall structure, lack of paragraphs, and lack of justifications/references for several claims overall leads me to believe that a significant amount of text could have been the output of an LLM.

My assessment is that the paper represents an initial step towards what should be a more comprehensive investigation, and currently, the paper lacks crucial justifications and has an undefined contribution – it neither advances performance for deep learning as claimed (since I believe the comparisons are not rigorous enough), nor does it yield much insight into neural computation. Overall, I believe that there are several critical weaknesses as discussed, which unfortunately make me strongly opposed to accepting the paper in its current form. I would encourage the authors to improve upon their work by considering these points and attempting to sharpen their work through more rigorous experimentation and better presentation.

**Questions:**

* Why is the performance of Hippo-medium quite consistently worse than Hippo-low?
* Could the authors comment on training times for Hippo-1 versus MLPs? The times reported for Hippo-1 are so much slower than the MLPs. Further, in some cases Hippo-high has been given more params than the MLP, but the performance is very close (e.g., CIFAR-10, CIFAR-100). Could the authors comment on this?
* Why do the authors consider a hippocampus-inspired architecture for visual tasks? In general, I feel there is a lack of justification for the model proposed here, and there are no references or explanations for why the authors consider the specific 5-way channel partition and implementation that they have explored here. Are there any references to prior work or indication that such computation is indeed performed in Hippocampal neurons?

---

### Official Review · Reviewer_vDKG · 2025-11-01

**Soundness:** 2
**Presentation:** 2
**Contribution:** 2
**Rating:** 2
**Confidence:** 4

**Summary:**

This paper introduces Hippo-1, a neural network architecture inspired by the CA3 region of the hippocampus, incorporating excitatory/inhibitory (E/I) populations and multiplicative shunting inhibition within multi-compartment neurons. The authors argue that this design preserves key dendritic integration principles while remaining computationally tractable. They report that Hippo-1 outperforms parameter-matched MLPs on several image-classification benchmarks (MNIST, CIFAR-10/100, Tiny-ImageNet), exhibits improved robustness to Gaussian and salt-and-pepper noise, and provides an accuracy gain per FLOP on harder datasets.

**Strengths:**

The paper tackles an interesting and timely problem: what advantages might biological structure confer on neural networks? Conceptually, the paper is interesting because it explores how dendritic-compartment interactions might improve learning performance and efficiency.

It is good to see an exploration of robustness and efficiency, in addition to just task performance. The experiments also cover a range of image datasets.

**Weaknesses:**

Hippo-1’s accuracy-per-FLOP is negative on simple datasets (MNIST, Fashion-MNIST) and only becomes positive where MLPs already perform poorly (CIFAR/Tiny-ImageNet). This suggests that the “improvement” largely reflects MLP inadequacy rather than a fundamental advance. The claim that the model scales better with complexity therefore lacks a solid baseline; comparisons with alternative architectures, such as convolutional connected networks for example would be more informative.

The paper does not articulate why or how the proposed architectural inductive biases drawn from biology benefit the tasks evaluated. The mechanisms drawn from hippocampal neurons, namely compartmentalized dendritic integration and multiplicative shunting inhibition between excitatory and inhibitory populations, are biologically well-motivated, but the authors do not analyze their computational role. The gains may just reflect a more expressive nonlinearity rather than a meaningful hippocampal computation. The authors could strengthen their case by performing ablations on shunting inhibition, E/I separation, or compartmental depth.

The paper is motivated by the hippocampal trisynaptic loop, in particular CA3’s role in it. Hippocampal CA3 is primarily associated with associative memory, pattern completion, and sequence learning, yet the evaluation focuses exclusively on image classification. The choice of image datasets is not justified and does not probe the motivating computational role of the architecture. A more appropriate test would involve memory-centric or reconstruction tasks. Relatedly, the architectural changes are primarily dendritic compartmentation. As such, the works’ motivation appears to be disconnected and it should focus more on dendritic computation.

The introduction has no citations.

The paper omits key references integrating dendritic computation and EI structure into ANNs ( Chavlis & Poirazi 2025, Poirazi & Mel 2001, Cornford 2021, Jones & Kording 2021 among others).

**Questions:**

See weaknesses. Minor points are:

- Can the multiplicative inhibition be negative, changing the polarity of the excitatory channel?
- Inhibition is only multiplicative here, why not also subtractive?
- What are the layers referred on line 235? There is only ever one, correct?
- The model description should be improved. There is also a tanh in figure 1 but not in section 3. Section 4 also introduces new parameters defining the model (d_out, hidden_dim). Why is n_stages even mentioned in table 1, it is always 1 and should be removed.

---

### Meta-Review · Area_Chair_c7EA · 2025-12-04

**Summary:**

The reviewers collectively recommended rejecting the paper due to flaws in experimental design, clarity, and theoretical motivation. A primary criticism was the inadequacy of the baselines; comparing the proposed Hippo-1 model only to simple, single-hidden-layer MLPs was viewed as insufficient, especially given that the reported accuracies are far below standard benchmarks for datasets like CIFAR-10. Reviewers highlighted unfair comparisons, noting that the proposed model benefited from extensive hyperparameter tuning and complex nonlinearities that were not afforded to the baselines, and they suggested comparisons against CNNs or other bio-inspired architectures would be necessary to validate the claims of the paper.

Furthermore, the reviewers questioned the fundamental premise of applying a hippocampal model, which is biologically associated with memory and sequence learning, to static image classification tasks without sufficient justification. They argued that this conceptual mismatch was compounded by presentation issues, particularly regarding the mathematical description of the model. Reviewers noted that the architecture is defined vaguely with missing weights and equations, making reproduction impossible, while the manuscript also suffers from garbled references and a lack of code. Ultimately, the consensus was that the paper requires more rigorous experimentation, a clearer model definition, and a more appropriate application domain to be considered for publication.

**Reviewer Concerns:**

The authors did not offer a rebuttal, so none of the concerns were addressed.

**Reviewer Scores:**

The original scores were 2, 2, 4. I do not think they would have changed.

---

### Decision · Program_Chairs · 2026-01-26

Reject